# Breed-Related Differential microRNA Expression and Analysis of Colostrum and Mature Milk Exosomes in Bamei and Landrace Pigs

**DOI:** 10.3390/ijms25010667

**Published:** 2024-01-04

**Authors:** Jie Li, Xuefeng Shang, Sen Zhang, Qiaoli Yang, Zunqiang Yan, Pengfei Wang, Xiaoli Gao, Shuangbao Gun, Xiaoyu Huang

**Affiliations:** 1College of Animal Science and Technology, Gansu Agricultural University, Lanzhou 730070, China; lijie5272@126.com (J.L.); 18809423537@163.com (X.S.); zhangsen6423@163.com (S.Z.); yangql0112@163.com (Q.Y.); yanzunqiang@163.com (Z.Y.); wangpf815@163.com (P.W.); gxl18892@163.com (X.G.); gunsbao056@126.com (S.G.); 2Gansu Research Center for Swine Production Engineering and Technology, Lanzhou 730070, China

**Keywords:** Bamei pigs, Landrace pigs, colostrum and mature milk, milk-derived exosomes, miRNA-seq

## Abstract

Breast milk, an indispensable source of immunological and nutrient components, is essential for the growth and development of newborn mammals. MicroRNAs (miRNAs) are present in various tissues and body fluids and are selectively packaged inside exosomes, a type of membrane vesicle. Milk exosomes have potential regulatory effects on the growth, development, and immunity of newborn piglets. To explore the differences in milk exosomes related to the breed and milk type, we isolated exosomes from colostrum and mature milk from domestic Bamei pigs and foreign Landrace pigs by using density gradient centrifugation and then characterized them by transmission electron microscopy (TEM) and nanoparticle tracking analysis (NTA). Furthermore, the profiles and functions of miRNAs in the two types of pig milk exosomes were investigated using miRNA-seq and bioinformatics analysis. We identified a total of 1081 known and 2311 novel miRNAs in pig milk exosomes from Bamei and Landrace pigs. These differentially expressed miRNAs (DE-miRNAs) are closely associated with processes such as cell signaling, cell physiology, and immune system development. Functional enrichment analysis showed that DE-miRNA target genes were significantly enriched in endocytosis, the T cell receptor signaling pathway, and the Th17 cell differentiation signaling pathway. The exosomal miRNAs in both the colostrum and mature milk of the two pig species showed significant differences. Based on related signaling pathways, we found that the colostrum of local pig breeds contained more immune-system-development-related miRNAs. This study provides new insights into the possible function of milk exosomal miRNAs in the development of the piglet immune system.

## 1. Introduction

The Bamei pig is one of the most famous local breeds in China, characterized by its strong resistance, roughage tolerance, superior reproduction ability, and excellent meat flavor [1,2]. Compared to foreign pig breeds introduced in China, such as Landrace and Yorkshire pigs, Bamei piglets exhibit stronger disease resistance, as evidenced by their lower rates of diarrhea, pathogen infection, and mortality. These qualities are not only related to genetic factors but also to the nutrition obtained during the lactation period. In recent years, with advances in genetic breeding techniques, sow litter size has increased significantly, and although improvements in sow reproductive performance have led to significant increases in lactation, piglet survival has declined [3].

The nutrient content of milk varies from animal to animal and changes dynamically over time [4,5]. The degree of variation in the content of milk protein, fat, lactose, minerals, and amino acids varies between breeds, and there are also large differences in the nutrient content of milk between strains [3]. Breast milk, an important source of passive immunity, is crucial for the growth and development of newborn piglets [6]. It contains many biologically active components that not only enforce passive immunity but also regulate the development of the immune system in breastfeeding infants [7]. Compared with foreign-introduced breeds, the Bamei pig has a lower milk yield, but the piglets are more resistant to adversity, which may be related to another active substance (exosomes) in the milk. Colostrum intake has been shown to markedly reduce mortality in pre-weaned piglets [8], which may be related to regulation by milk exosomal miRNAs [6].

Exosomes are membranous vesicles secreted by living cells, with diameters of approximately 30–150 nm [9]. They are widely distributed in body fluids, such as breast milk, saliva, blood, and urine, and can carry proteins, mRNAs, and miRNAs that participate in intercellular communication [3,10,11] and immune responses [12,13], and they have been used for drug delivery [14]. MiRNAs in breast milk exosomes remain stable under acidic conditions and are absorbed by the infant’s intestine to perform various regulatory functions [15]. Exosomes have been shown to attenuate cellular damage from *deoxynivalenol* by regulating miRNA expression in IPEC-J2 cells [16] and promoting intestinal cell proliferation and development [17]. Also, exosomes can attenuate lipopolysaccharide (LPS)-induced apoptosis by inhibiting TLR4/NF-κB and p53 pathways in intestinal epithelial cells [18]. Lin et al. [19] reported that oral ingestion of bovine and pig milk exosomes in piglets increased the serum levels of miRNAs. Bovine-milk-derived exosomes contain immunomodulatory miRNAs [7]. A previous study showed that pig milk exosomes contain immune-related miRNAs, and the miRNAs in colostrum are higher than those in mature milk [20], which can regulate the body’s immune response [21]. Therefore, it can be hypothesized that milk-derived exosomal miRNAs are critical for the growth and development of the immune system in neonatal animals.

MiRNAs are conserved small non-coding nucleotide molecules of about 18–25 nt in length [22]. Several studies have demonstrated that miRNAs regulate apoptosis, proliferation, differentiation, immunity, growth, development, host immunity, and other physiological processes [23,24,25,26]. MiRNAs can be stably present in body fluids, including saliva, urine, breast milk, and blood [27]. Exosomal miRNAs have been receiving great attention from researchers for their intercellular messaging role.

So far, only a few studies have examined the miRNA expression profiles of milk exosomes from different breeds within a mammalian species. For instance, Özdemir et al. [28] studied colostrum and mature milk miRNAs in two different bovine species and identified 545 known miRNAs and 260 novel miRNAs. Izumi et al. [29] detected 100 and 53 miRNAs in bovine colostrum and mature milk, respectively, and found that they were mostly related to immune system development. Although differentially expressed miRNAs (DE-miRNAs) have been identified between pig colostrum and mature milk [20], identifying DE-miRNAs from different pig breeds in different lactation periods has not been attempted so far and can reveal interesting regulatory information. Accordingly, in this study, we examined milk exosomes to identify miRNA characters in colostrum and mature milk from two different breeds of pigs, Bamei and Landrace, as well as the potential functions of the identified DE-miRNAs and related target genes. Our results lay the foundation for further functional studies on milk-derived exosomal miRNAs.

## 2. Results

### 2.1. Exosome Characterization

Through extraction and identification, TEM revealed that the milk exosomes from Bamei and Landrace pigs were round or oval (Figure 1(A1–A4)), and their main particle sizes ranged from 50 to 100 nm (Figure 1(B1–B4)). There was no significant difference in exosome particle size between the colostrum and mature milk of Bamei and Landrace pigs (Figure 1C). The colostrum exosome concentration was extremely significantly higher in both Bamei and Landrace pigs than in mature milk (*p* < 0.01), the colostrum exosome concentration was extremely significantly higher in Landrace pigs than in Bamei pig colostrum (*p* < 0.01), and the mature milk exosome concentration was extremely significantly higher in Landrace pigs than in Bamei pig mature milk (*p* < 0.01) (Figure 1D). The above results showed that the isolated exosomes had good morphological characteristics and high concentrations, which satisfied the research requirements.

### 2.2. Quality Control of Sequencing Data

To identify exosomal miRNAs in the colostrum and mature milk of Bamei and Landrace pigs, 12 small RNA libraries were constructed and sequenced, namely, the BMo *(n* = 3), BMc (*n* = 3), LWo (*n* = 3), and LWc (*n* = 3) groups, respectively (Section 4.2). In total, 108,497,788 (valid reads 25,444,101), 112,049,590 (valid reads 32,228,194), 108,283,662 (valid reads 14,426,719), and 104,916,517 (valid reads 31,565,425) total reads were obtained in the BMo, BMc, LWo, and LWc groups, respectively (Table 1).

We performed a length distribution statistical analysis on the total number of filtered miRNA data. The length distribution of exosomal miRNAs was similar between Bamei and Landrace pigs; most of the miRNAs were in the range of 18–22 nt in length (Appendix A).

### 2.3. miRNA Identification and Prediction Analysis

In total, 1081 known and 2311 novel miRNAs were identified from the twelve milk samples (Appendix A). There were 37, 81, 9, and 49 specific miRNAs in BMo, BMc, LWo, and LWc, respectively. There were 362 miRNAs identified in common between BMc and BMo, 223 miRNAs were common between LWc and LWo, 254 miRNAs were common between BMo and LWo, and 421 miRNAs were common between BMc and LWc (Figure 2A). In addition, there were 203 common miRNAs among the four groups. Among these, 5 miRNAs (ssc-miR-30a-5p, ssc-miR-21-5p, mmu-miR-5124a, cfa-miR-200c, and ssc-miR-148a-3p) were differentially expressed among the top 10 most abundantly expressed miRNAs (Figure 2B). We further analyzed the chromosomal distribution of the identified miRNAs and found that the miRNAs in the four groups were distributed on all chromosomes, with the highest distribution on chromosome one (Appendix A).

### 2.4. Identification of DE-miRNAs

A total of 52 (30 upregulated and 22 downregulated), 74 (61 upregulated and 13 downregulated), 18 (12 upregulated and 6 downregulated), and 52 (18 upregulated and 34 downregulated) DE-miRNAs were identified from the BMc vs. BMo, LWc vs. LWo, BMo vs. LWo, and BMc vs. LWc comparisons, respectively (Figure 3A, Appendix A); in total, 137 DE-miRNAs were identified in the four comparison groups (Figure 3B). Furthermore, we list some of the miRNAs with significantly high expression in different comparison groups (Table 2). Volcano plots depicting the overall distribution of DE-miRNAs among the four comparison groups are shown in Figure 3C–F. Concisely, the above study showed that miRNAs were differentially expressed in pig colostrum and mature milk exosomes.

### 2.5. Target Gene Prediction and Functional Enrichment Analysis of DE-miRNAs

To better understand the functions of DE-miRNAs in milk exosomes, the target genes of DE-miRNAs were predicted using TargetScan (v5.0) and miRanda (v3.3a) software. In total, 12,291 gene targets were predicted for the 52 DE-miRNAs between BMc and BMo (Appendix A); 12,727 gene targets were predicted for the 74 DE-miRNAs between LWc and LWo (Appendix A); 9555 gene targets were predicted for the 18 DE-miRNAs between BMo and LWo (Appendix A); and 12,166 gene targets were predicted for the 52 DE-miRNAs between BMc and LWc (Appendix A).

Next, we performed functional enrichment analysis on the identified DE-miRNA target genes. In total, 881, 848, 654, and 839 significantly enriched GO terms were identified in BMc vs. BMo, LWc vs. LWo, BMo vs. LWo, and BMc vs. LWc comparison groups, respectively (*p* < 0.01; Appendix A). GO and KEGG enrichment results were essentially the same in the four comparison groups, where regulation of transcription, DNA-templated (GO:0006355), positive regulation of transcription by RNA polymerase II (GO:0045944), and signal transduction (GO:0007165) are the terms with the highest number of enriched genes in the biological process category. Membrane (GO:0016020), integral component of membrane (GO:0016021), and nucleus (GO:0005634) are the top three terms in the cellular component category, while protein binding (GO:0005515), metal ion binding (GO:0046872), and ATP binding (GO:0005524) are the top terms in the molecular function category (Figure 4A–D). The identified DE-miRNA target genes are involved in immune regulation, cell differentiation, and other biological processes. Concerning KEGG pathway analysis, DE-mRNAs in the BMc vs. BMo, LWc vs. LWo, BMo vs. LWo, and BMc vs. LWc comparison groups were mainly assigned to 146, 129, 109, and 137 KEGG pathways, respectively (*p* < 0.01; Appendix A). The enrichment of DE-miRNA target gene signaling pathways was similar in the BMc vs. BMo and LWc vs. LWo groups, including endocytosis (ko04144), T cell receptor signaling pathway (ko04660), human T-cell leukemia virus 1 infection (ko05166), and TNF signaling pathway (ko04668). In BMo vs. LWo, DE-miRNA targets were found to be linked to autophagy-animal (ko04140), Th17 cell differentiation (ko04659), AMPK signaling pathway (ko04152), and MAPK signaling pathway (ko04010), and other signaling pathways were also significantly enriched (Figure 5A–D; Appendix A).

Overall, these results indicate that DE-miRNAs in the milk exosomes of Bamei and Landrace pigs are mainly involved in immunity regulation, autoimmune diseases, and the development of the immune system.

### 2.6. miRNA-mRNA Interaction Network Analysis

To investigate the key mRNA interactions of milk exosome DE-miRNAs, we predicted the target genes of the highest upregulated miRNAs (ssc-miR-551a, ssc-miR-205, ssc-miR-7137-5p, and ssc-miR-29c) and the highest downregulated miRNAs (ssc-miR-143-5p, ssc-miR-7-3p, ssc-miR-193a-3p, and ssc-miR-4335-p5) selected from the four comparison groups. Some of these identified miRNA-mRNA relationship pairs were used to construct the interaction network (Figure 6A). The eight selected miRNA target genes were enriched in 511 GO terms and 82 KEGG pathways (*p* < 0.01; Appendix A); the significantly enriched signaling pathways were associated with tumorigenesis, cell signaling, and innate immune response. These results suggest that mRNA–miRNA interaction may be a key regulator of the development of the host immune system.

From the screening of target gene enrichment analysis, we selected the immune-related pathways, including the T cell receptor signaling pathway, human T-cell leukemia virus 1 infection, and Th17 cell differentiation, as nodes to collate the pathways associated with the miRNA-mRNA interaction network. Network mapping with Cytoscape 3.8.0 software (Figure 6B) suggested that milk exosomal miRNAs may regulate the development of the immune system by targeting mRNAs.

### 2.7. Conservation of the Identified miRNAs in Other Species

We next analyzed the conservation of pig milk exosomal miRNAs in other species using number counts and histograms for the frequency of miRNAs (Figure 7). We found that pig milk exosomal miRNAs are highly conserved in *Homo sapiens* (hsa), *Bos taurus* (bta), *Sus scrofa* (ssc), *Mus musculus* (mmu), *Equus caballus* (eca), and *Pan troglodytes* (ptr) while showing lower conservation in *Pygathrix bieti* (pbi).

### 2.8. RT-qPCR Validation of DE-miRNAs

To verify the accuracy of the sequencing results, we randomly selected 12 DE-miRNAs for RT-qPCR validation. ssc-miR-205, ssc-miR-1296-5p, and ssc-miR-455-5p were upregulated and ssc-miR-29c, ssc-miR-21-5p, and ssc-miR-29a-3p were downregulated in Bamei pig mature milk exosomes compared to in colostrum (Figure 8A). Likewise, ssc-miR-128, ssc-miR-221-3p, and ssc-miR-222 were upregulated and ssc-miR-23a, ssc-miR-22-3p, and ssc-miR-7-3p were downregulated in Landrace pigs’ mature milk exosomes compared to in colostrum (Figure 8B). The expression patterns of these miRNAs detected by RT-qPCR were consistent with the miRNA-seq data, indicating that the sequencing results are reliable.

## 3. Discussion

Exosomes are cell-derived vesicles that are abundant in breast milk and participate in intercellular signaling and inflammatory and immune responses [30]. MiRNAs play a critical role in the development of the immune system. Recently, pig milk exosomal miRNAs have become a hot research topic. Chen et al. [31] sequenced pig milk exosomes and found that they contained 176 known and 315 novel mature miRNAs. Liu et al. [32] detected 208 pre-miRNAs and 200 mature miRNAs in pig milk exosomes. However, it is not known whether the characteristics of milk exosomes and miRNA expression are related to the breed and milk type.

Here, we characterized the colostrum and mature milk exosomes of both Bamei and Landrace pigs using TEM and NTA. We found that colostrum and mature milk exosomes of both Bamei and Landrace pigs were round or oval in shape, with no significant difference in particle size, and the main particle size was 50–100 nm, which is consistent with the previously reported exosome morphology [33]. Whether it was Bamei or Landrace pigs, the concentration of colostrum exosomes was significantly higher than that of mature milk exosomes. This may be due to the large differences in the different nutrient requirements of piglets at different stages of lactation. Then, we systematically reported the expression profiles of miRNAs in the colostrum and mature milk exosomes of Bamei and Landrace pigs. There were 1081 known miRNAs and 2311 new miRNAs identified from the four comparison groups. Our study revealed more pig milk exosomal miRNAs. The majority of miRNAs were 18–22 nt in length, which is consistent with the results reported in previous studies [31]. We found that five of the most abundantly expressed miRNAs, ssc-miR-30a-5p, ssc-miR-21-5p, mmu-miR-5124a, cfa-miR-200c, and ssc-miR-148a-3p, were common between the colostrum and mature milk in both pigs. Consistent with our study, Zhang et al. [33] extracted exosomes from pig urine, plasma, semen, and bile and identified miR-148a-3p, miR-21-5p, and so on, which regulate the development of the immune system. Interestingly, ssc-miR-21-5p, mmu-miR-5124a, and ssc-miR-148a-3p were more abundant in colostrum exosomes than in mature milk exosomes in both the Bamei and Landrace pigs, whereas ssc-miR-30a-5p was less abundant in colostrum exosomes than in mature milk exosomes; these may be key miRNAs for different milk types. It was demonstrated that miR-21-5p was involved in the LPS-induced inflammatory response in H9c2 cells, and overexpression inhibited the inflammatory response [34]. Another study demonstrated that exosomal miR-21-5p can regulate the migration and proliferation of cancer cells [35]. In addition, Zhou et al. [36] found that the milk-derived exosomal miR-148a-3p was highly expressed in milk and was associated with immunity.

In order to further analyze the differential miRNAs in colostrum and mature milk exosomes of Bamei and Landrace pigs, we analyzed the differential expression of exosomal miRNAs between different milk types and different breeds, respectively. Between the different milk types, ssc-miR-340 and ssc-miR-30e-5p were significantly upregulated and highly expressed in the mature milk of Bamei pigs compared to the colostrum of Bamei pigs; in addition, we found that most miRNAs were upregulated in the colostrum of the Bamei pigs, and some miRNAs with higher expression in the colostrum of Bamei pigs were related to the development of the immune system, such as ssc-miR-143-5p, ssc-miR-7137-5p, and hsa-miR-3195. These may contribute to the high disease resistance in Bamei piglets. Compared with Landrace pig colostrum, ssc-miR-205 and ssc-miR-221-3p were significantly upregulated and highly expressed in Landrace pig mature milk. Between the different breeds, ssc-miR-29a-3p and ssc-miR-103 were significantly upregulated and highly expressed in the mature milk of Bamei pigs compared to the mature milk of Landrace pigs, and ssc-miR-141 and ssc-miR-22-3p were significantly upregulated and highly expressed in the colostrum of Landrace pigs compared to the colostrum of Bamei pigs. The reason for the differences in the expression of exosomal miRNA in colostrum and mature milk between the different breeds may be due to the large differences in the developmental rates of the compared breeds [37]. In addition, the trait differences between different breeds are unique physiological traits formed through long-term natural and artificial selection, which are affected by the regulation of different gene expression patterns and show different phenotypes. Therefore, there are differences in gene expression and transcriptional regulation between different breeds at different stages of their growth, development, and physiology. The colostrum and mature milk exosomes of the Bamei and Landrace pigs have indeed shown marked differences in adaptation to their respective physiological characteristics and breed characteristics over a long evolutionary period. The exosomal miRNAs were significantly different in both the colostrum and mature milk between the two breeds, and the exosomes of domestic pig milk contained richer immune-related miRNAs.

In this study, we performed a functional enrichment analysis of target genes of DE- miRNAs in four comparative groups. GO enrichment analysis showed that GO terms were most significantly enriched in the cytosol, cytoplasm, and nucleus terms annotated as cellular components. Concerning the KEGG pathway analysis, we found that DE-miRNA target genes were involved in processes such as cell signaling, cell physiology, and immune system development, and significantly enriched pathways were T cell receptor signaling pathway, human T-cell leukemia virus 1 infection, Th17 cell differentiation, and AMPK signaling pathway. The T cell receptor signaling pathway plays an essential role in innate and adaptive immunity [38]. It was shown that miRNAs can regulate immune inflammatory responses by inhibiting or activating the T cell receptor signaling pathway [39]. In addition, the T cell receptor signaling pathway is important for T -cell development, and the breakdown of this pathway leads to autoimmunity [40]. Several studies have shown that Th17 cell differentiation is closely associated with the development of many autoimmune and inflammatory diseases [41,42]. AMPK, an essential kinase involved in energy homeostasis, is a key signal transduction protein [43]. The AMPK signaling pathway has an important regulatory function in the regulation of metabolism and cell growth [44]. The endocytosis pathway was most significantly enriched in the BMc vs. BMo, LWc vs. LWo, and BMc vs. LWc comparison groups. Endocytosis is a way of signaling that is important in the regulation of nutrient internalization and signal transduction [45,46]. In contrast, the autophagy-animal significance was highest in the BMo vs. LWo group. Autophagy helps maintain cellular homeostasis [47] and plays a signal transduction role in the immune system [48]. Elevated or impaired levels of autophagy have great significance in cancer, metabolic, and heart diseases [49,50]. This shows that differentially expressed miRNA target genes are significantly enriched in signaling pathways related to immunity and growth and development and may play a role in regulating the development of the immune system, with the degree of enrichment related to the breed and milk type.

To further screen key candidate miRNAs related to the breed and milk type, we constructed miRNA-mRNA and miRNA-miRNA-KEGG relational networks by conducting bioinformatics analyses to predict key miRNAs and network pathways regulating the immune system. In the interactional network, ssc-miR-29c, ssc-miR-7-3p, and ssc-miR-193a-3p are involved in immune signaling pathways, such as the AMPK signaling pathway and Th17 cell differentiation. These miRNAs appear to activate genes associated with cell signaling and immune responses (such as *IL6*, *TNF*, and *STAT6*) that play an active role in stimulating the maturation of the immune system. Although our study has not yet demonstrated that milk exosomal miRNAs promote immune system development by targeting mRNAs, we will attempt to confirm this experimentally in the future.

A previous study revealed that milk-derived miRNAs are conserved among mammals [51]. By performing a conservation analysis, we found that the colostrum and mature milk exosomal miRNAs of the Bamei and Landrace pigs were highly conserved among different species. The let-7 family miRNA was the most conserved and was expressed in all four comparison groups. Studies have shown that let-7 miRNAs regulate immune responses [52] and other cellular processes [53], and the overexpression of let-7d inhibited the release of inflammatory factors and, in turn, suppressed the inflammatory response in neonatal rats with necrotizing enterocolitis [54]. The let-7 family miRNAs are the key regulators of animal growth and development [36]. The expression of let-7 in all four comparison groups suggests that piglets may obtain milk exosomal miRNAs through intestinal absorption by sucking pig’s milk, thus obtaining abundant nutrition to promote their own growth and development and improve the immune system [19,20,36]. However, whether it can be detected in piglets needs to be experimentally verified.

## 4. Materials and Methods

### 4.1. Experimental Animal Selection

This study was carried out on the breeding farm of Pingliang, Gansu, China. Three sows from each species, Bamei and Landrace, in good health and with the same body condition, with a similar litter size, breeding period, and farrowing period, were selected. They were transferred to the farrowing house seven days before farrowing to adapt to the pen environment and kept under the same environmental conditions, with natural light and free access to food and water. The sows were fed according to the National Research Council (NRC) (2004) for the nutritional level of sows during lactation. All experimental pigs were immunized according to the routine immunization program, and the barn was disinfected once a week.

### 4.2. Milk Collection and Exosome Extraction

In total, 12 sequencing samples were collected from Bamei pigs’ colostrum (BMo) and mature milk (BMc) and Landrace pigs’ colostrum (LWo) and mature milk (LWc) at one and seven days postpartum (3 samples from each group), respectively.

Milk exosomes were extracted by density gradient centrifugation. Briefly, the milk samples were rapidly dissolved in a 37 °C water bath and then centrifuged at 2000× *g*, followed by another centrifugation at 10,000× *g* for 45 min at 4 °C to remove the larger vesicles. The obtained supernatant was centrifuged in an over-speed rotor (CP100MX, Hitachi, Tokyo, Japan) at 100,000× *g* and 4 °C for 120 min. The supernatant was discarded. The precipitate was resuspended in 20 mL of pre-chilled 1 × PBS (phosphate-buffered saline), followed by centrifugation at 2000× *g* for 30 min at 4 °C. The obtained supernatant was again centrifuged at 100,000× *g* for 120 min at 4 °C. The supernatant was removed, and the precipitate was resuspended in 1 mL of pre-chilled PBS and temporarily stored at 4 °C. To prepare 40, 20, 10, and 5% iodixanol mixtures, different concentrations of iodixanol solutions were added successively along the wall of the tube into the super-isolation tube. Finally, 1 mL of the resuspension stored at 4 °C was added to the top layer and centrifuged at 100,000× *g* for 120 min at 4 °C. After centrifugation, the liquid was divided into 12 layers, and the liquid in the middle 6–9 layers was removed and then centrifuged at 4 °C for 120 min at 100,000× *g*. The supernatant was discarded, and the pellet was resuspended in 200 μL of pre-chilled 1 × PBS for immediate detection. Finally, the extracted exosomes were purified according to the instructions of the Cell Culture Media Exosome Purification Kits (cat. 60400, NORGEN, Thorold, ON, Canada).

### 4.3. Characterization of Milk Exosomes

First, 10 μL of each exosome sample was aspirated and added dropwise onto copper grids to precipitate for 1 min, and the floating liquid was removed with filter paper. Then, 10 μL drops of uranyl acetate were added to the copper grids to precipitate for 1 min and cleaned with ddH_2_O. After drying at room temperature for several minutes, imaging was performed using transmission electron microscopy (TEM, cat. HT-7700, Hitachi, Tokyo, Japan). The particle size and concentration information were detected using a Nanoflow detector (NanoFCM, N30E, Xiamen, China).

### 4.4. Library Construction and Sequencing

The total RNA from milk exosomes was extracted from the 12 samples (Section 4.2) using the Exosomal RNA Isolation Kit (cat. NGB-58000, NORGEN, Thorold, ON, Canada) according to the manufacturer’s instructions. The purity and integrity of RNA were assessed using a Nanodrop 8000 spectrophotometer (Nanodrop Technologies, Wilmington, DE, USA) and an Agilent 2100 Bioanalyzer (Agilent Technologies, Santa Clara, CA, USA), respectively. The RNA samples with RIN ≥ 7 were used for the construction of miRNA libraries using the TruSeq Small RNA Sample Prep Kits (Illumina, San Diego, CA, USA). The constructed libraries were sequenced using the Illumina Hiseq 2500 platform (LC Bio-Technology CO., Ltd., Hangzhou, China) with a single-end 1 × 50 bp strategy.

### 4.5. Raw Data Quality Control

The raw data were analyzed using ACGT101-miR (v4.2, LC Sciences, Houston, TX, USA) software, and the analysis process was as follows: the raw data were quality-controlled to obtain clean reads by removing 3’ adapter sequences, and the length of the retained sequences was between 18 and 26 nt. The reads after filtering for length were localized to the pig reference genome sequence (*Sus scrofa* 11.1) using Bowtie [55]. The remaining sequences were matched to the mRNA (ftp://ftp.ensembl.org/pub/release-96/fasta/sus_scrofa/dna/) (accessed on 12 April 2023), Rfam (http://rfam.janelia.org) (accessed on 12 April 2023), and Repbase databases (http://www.girinst.org/education/index.html) (accessed on 12 April 2023). The final obtained data were the valid reads, which were used for subsequent bioinformatics analysis.

### 4.6. Bioinformatics Analysis

Differential miRNA analysis was performed using the R package DESeq2 (v1.22.2) software [56], and the screening condition for DE-miRNAs was set to a *p* ≤ 0.05. Two computational target prediction algorithms, TargetScan (v5.0) [57,58,59] and miRanda (v3.3a) [60,61,62], were used to predict the target genes of milk exosomal miRNAs; TargetScan_score ≥ 50 and miRanda_Energy < -10 were the screening conditions, respectively. Sequences at the intersection of the two software programs were considered the final target genes of the DE-miRNAs. The targeted genes of miRNAs were subjected to Gene Ontology (GO) and Kyoto Encyclopedia of Genes and Genomes (KEGG) pathway enrichment analysis using the DAVID bioinformatic resource [63]. Cytoscape (v3.8.0) was used to map miRNA-mRNA and miRNA-miRNA-KEGG interaction networks. The VennDiagram package (v1.6.20) was used to draw Venn diagrams. Advanced volcano plots were generated using OmicStudio (v1.0.3) tools (https://www.omicstudio.cn/tool/7) (accessed on 18 May 2023).

### 4.7. RT-qPCR Validation of Differentially Expressed Genes

The synthesis of cDNA for miRNA quantification was performed using the Mir-X™ miRNA First-Strand Synthesis Kit (Takara, Dalian, China) as per the manufacturer’s instructions. RT-qPCR was performed using the Roche LightCycler 480 II platform (Roche, Penzberg, Germany) using 2 × SYBR^®^ Green ProTaq HS Premix II (Accurate Biotechnology, Changsha, China). All experiments were repeated three times, and relative gene expression was calculated using the 2^−∆∆Ct^ method [64]. U6 was the endogenous control. The used primers’ information is listed in Table 3.

### 4.8. Statistical Analysis

The IBM SPSS 21.0 software (IBM Corp., Chicago, IL, USA) was used for statistical analysis of the data. Comparisons between the two groups were performed with the T-test. Experimental data are displayed as mean ± standard deviation (SD). Plotting was performed using GraphPad Prism 8.0 (GraphPad Inc., La Jolla, CA, USA). *p* < 0.05 indicates significant differences, and *p* < 0.01 indicates extremely significant differences.

## 5. Conclusions

In summary, there were no differences in the morphology and particle size of pig milk exosomes between different breeds, but there were differences in concentration. We identified 1081 known and 2311 new miRNAs in milk exosomes from Bamei and Landrace pigs. The miRNA expression in pig milk exosomes varies with the breed and milk type. Bamei pig colostrum is richer in immune-related miRNAs, and these miRNAs may regulate the immune and growth performance of Bamei piglets by activating immune- and growth-development-related genes or pathways, and their roles need to be confirmed by further experiments.

## Figures and Tables

**Figure 1 ijms-25-00667-f001:**
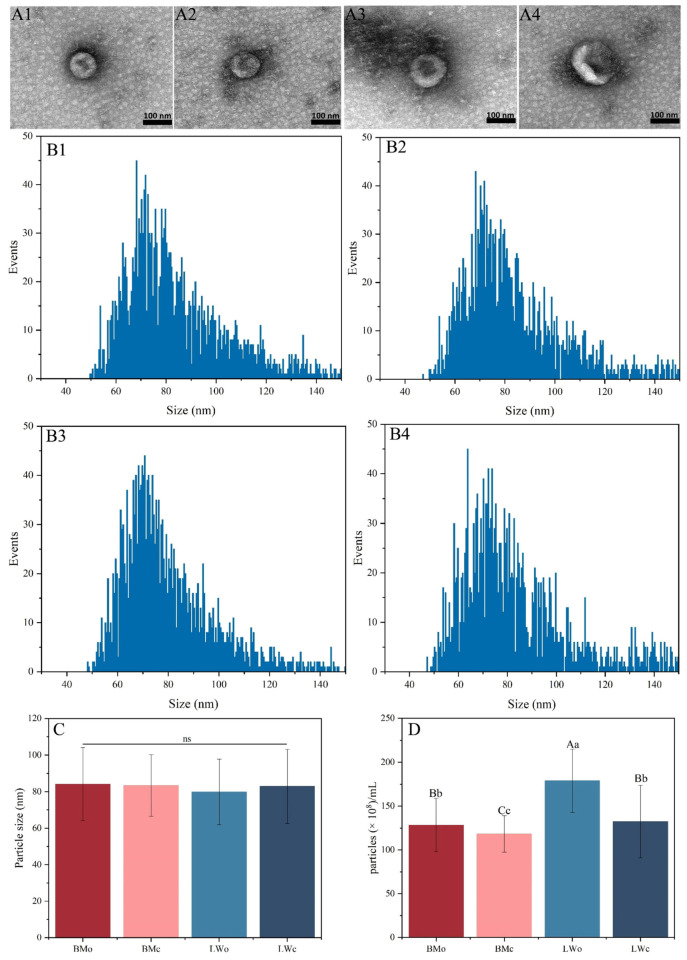
The shape, particle size, and concentration analysis of pig milk exosomes. Transmission electron microscopy (TEM) images of colostrum (**A1**) and mature milk exosomes (**A2**) of Bamei pigs (scale bar 100 nm). TEM images of colostrum (**A3**) and mature milk exosomes (**A4**) of Landrace pigs (scale bar 100 nm). Statistical graph of particle size distribution of colostrum (**B1**) and mature milk exosomes (**B2**) of Bamei pigs. Statistical graph of particle size distribution of colostrum (**B3**) and mature milk exosomes (**B4**) of Landrace pigs. (**C**) The average particle size of exosomes. (**D**) Concentration of the milk exosomes of Bamei and Landrace pigs. BMo means Bamei pigs’ colostrum exosomes, BMc means Bamei pigs’ mature milk exosomes, LWo means Landrace pigs’ colostrum exosomes, and LWc means Landrace pigs’ mature milk exosomes. Different capital letters indicate extremely significant differences (*p* < 0.01). Different lowercase letters indicate significant differences (*p* < 0.05); ns, the same capital or lowercase letters indicate non-significant differences.

**Figure 2 ijms-25-00667-f002:**
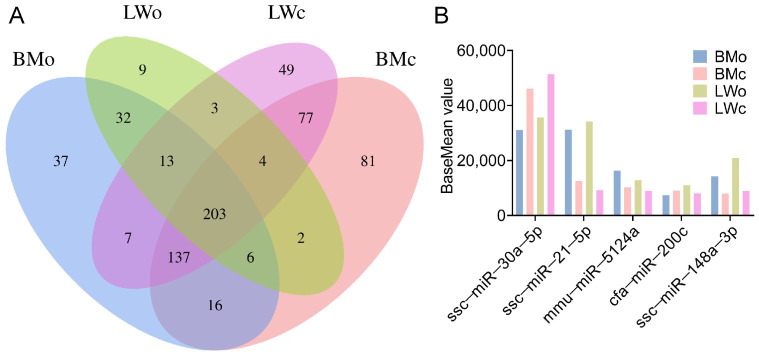
Identification and predictive analysis of pig milk exosomal miRNAs. (**A**) Venn diagram showing the number of total miRNAs identified in BMo, BMc, LWo, and LWc samples. (**B**) The five most abundant DE-miRNAs.

**Figure 3 ijms-25-00667-f003:**
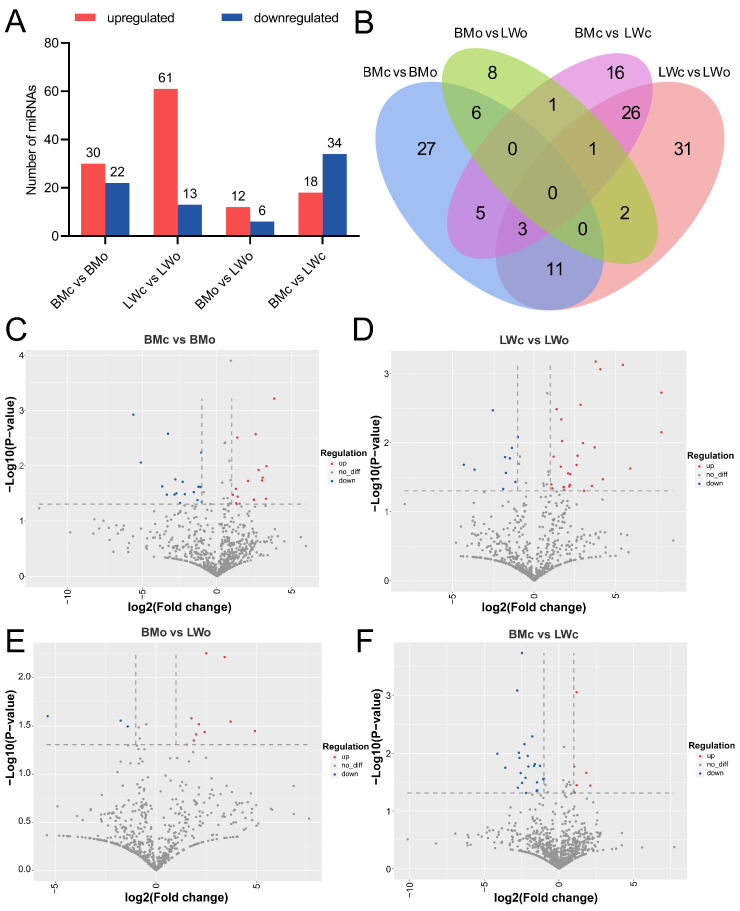
Differential analysis of DE-miRNAs in pig milk exosomes. (**A**) Distribution of up- and downregulated DE-miRNAs in respective comparison groups. (**B**) Venn diagram and (**C**–**F**) volcano plots of DE-miRNAs in four comparison groups.

**Figure 4 ijms-25-00667-f004:**
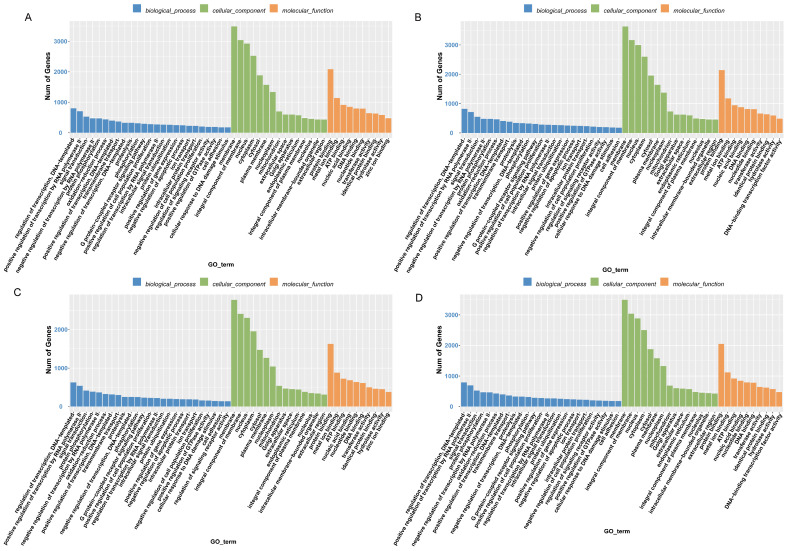
The GO enrichment histogram of DE-miRNA target genes. (**A**) BMc vs. BMo. (**B**) LWc vs. LWo. (**C**) BMo vs. LWo. (**D**) BMc vs. LWc.

**Figure 5 ijms-25-00667-f005:**
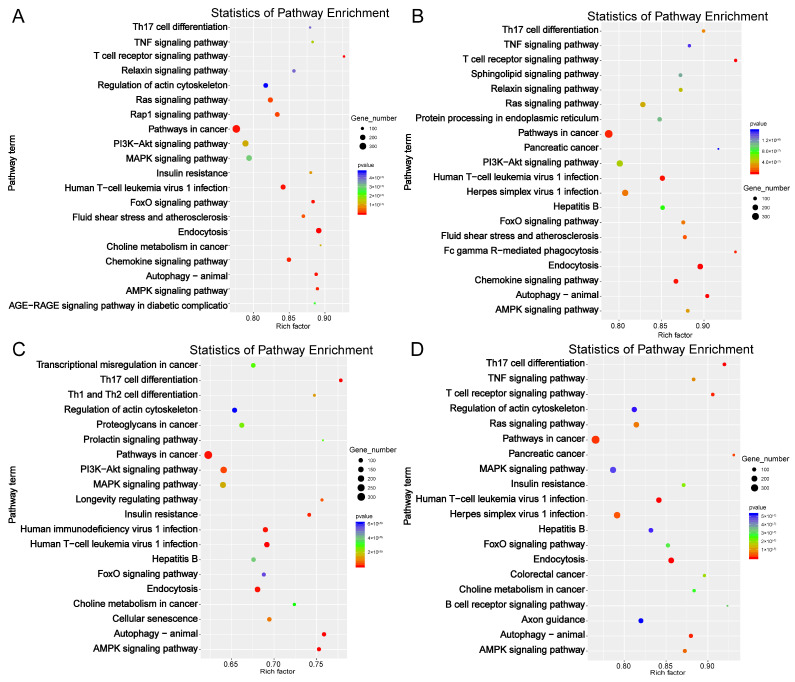
The KEGG enrichment scatter plots of DE-miRNA target genes. (**A**) BMc vs. BMo. (**B**) LWc vs. LWo. (**C**) BMo vs. LWo. (**D**) BMc vs. LWc.

**Figure 6 ijms-25-00667-f006:**
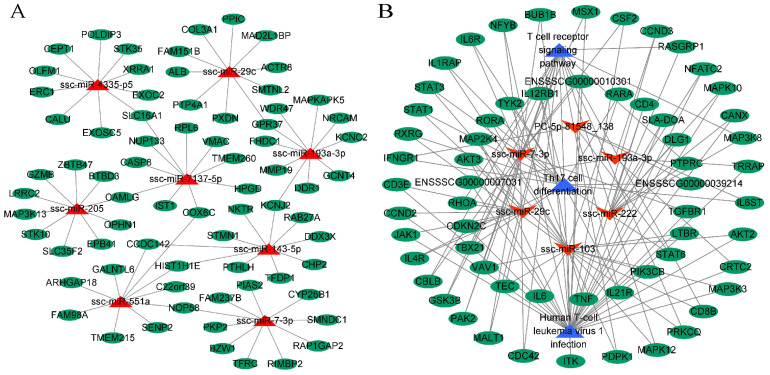
miRNA interaction network analysis. (**A**) miRNA-mRNA interaction network of DE-miRNAs and their corresponding target genes. (**B**) miRNA-mRNA-KEGG network diagram. miRNAs are in red, mRNAs are in green, and KEGG pathways are in blue.

**Figure 7 ijms-25-00667-f007:**
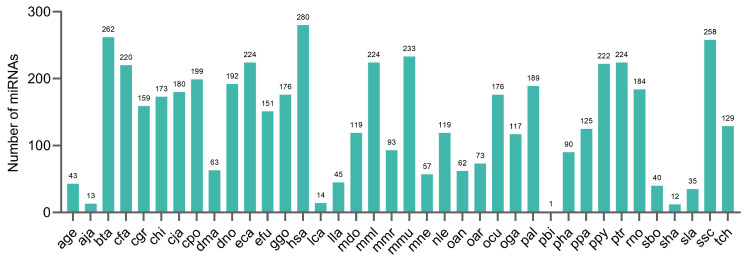
Conservation analysis of pig milk exosomal miRNAs in other species. The horizontal coordinates indicate different species, and the vertical coordinates indicate the occurrences of the marked precursor in that species.

**Figure 8 ijms-25-00667-f008:**
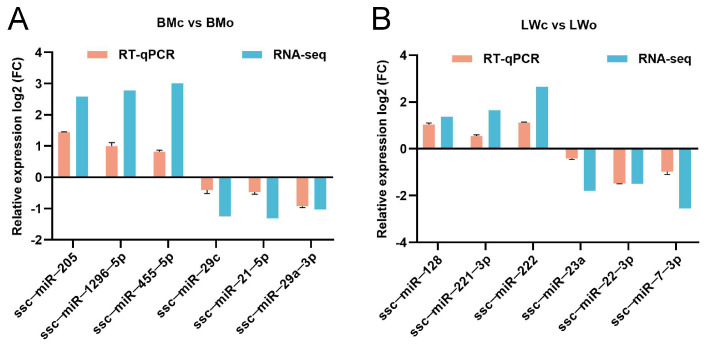
Validation of DE-miRNAs by RT-qPCR. DE-miRNAs in (**A**) BMc vs. BMo and (**B**) LWc vs. LWo.

**Table 1 ijms-25-00667-t001:** Sequencing data quality control.

Sample	Total Reads	Total Bases	Valid Reads	Valid Ratio%	Q20%	Q30%	GC%
BMo-1	38,893,587	1,983,572,937	9,441,552	24.28	98.06	94.19	57.34
BMo-2	35,837,285	1,827,701,535	7,261,431	20.26	97.98	94.08	56.19
BMo-3	33,766,916	1,722,112,716	8,741,118	25.89	97.93	94.08	57.40
BMc-1	40,678,884	2,074,623,084	11,340,306	27.88	98.08	94.34	54.35
BMc-2	40,108,481	2,045,532,531	11,316,963	28.22	98.05	94.30	56.13
BMc-3	31,262,225	1,594,373,475	9,570,925	30.61	97.70	93.30	55.17
LWo-1	36,276,785	1,850,116,035	6,949,261	19.16	97.51	93.03	57.01
LWo-2	38,368,116	1,956,773,916	4,574,542	11.92	98.07	94.33	53.28
LWo-3	33,638,761	1,715,576,811	2,902,916	8.63	97.73	93.55	57.57
LWc-1	33,048,336	1,685,465,136	11,280,944	34.13	98.11	94.34	54.90
LWc-2	39,577,542	2,018,454,642	10,166,522	25.69	98.15	94.5	55.78
LWc-3	32,290,639	1,646,822,589	10,117,959	31.33	98.11	94.44	56.05

Note: BMo means Bamei pigs’ colostrum exosomes, BMc means Bamei pigs’ mature milk exosomes, LWo means Landrace pigs’ colostrum exosomes, and LWc means Landrace pigs’ mature milk exosomes.

**Table 2 ijms-25-00667-t002:** Some of the significantly highly expressed miRNAs in the four comparison groups.

miRNA Name	miRNA Sequence (5′-3′)	Groups	Up/Down
ssc-miR-340	TTATAAAGCAATGAGACTGATT	BMc vs. BMo	up
ssc-miR-30e-5p	TGTAAACATCCTTGACTGGAAGCT	BMc vs. BMo	up
ssc-miR-205	TCCTTCATTCCACCGGAGTCTGT	LWc vs. LWo	up
ssc-miR-221-3p	AGCTACATTGTCTGCTGGGTTT	LWc vs. LWo	up
ssc-miR-29a-3p	TAGCACCATCTGAAATCGGTTA	BMc vs. LWc	up
ssc-miR-103	AGCAGCATTGTACAGGGCTATGA	BMc vs. LWc	up
ssc-miR-141	TAACACTGTCTGGTAAAGATGGC	BMo vs. LWo	down
ssc-miR-22-3p	AAGCTGCCAGTTGAAGAACTGT	BMo vs. LWo	down

**Table 3 ijms-25-00667-t003:** Primers of miRNAs for RT-qPCR.

miRNA Name	Primer Sequence (5′-3′)
ssc-miR-205	TCCTTCATTCCACCGGAGTCT
ssc-miR-1296-5p	TTAGGGCCCTGGCTCCATCT
ssc-miR-455-5p	TGTGCCTTTGGACTACATCG
ssc-miR-29c	TAGCACCATTTGAAATCGG
ssc-miR-21-5p	GCTTATCAGACTGATGTTG
ssc-miR-29a-3p	TAGCACCATCTGAAATCGGT
ssc-miR-128	TCACAGTGAACCGGTCTCT
ssc-miR-221-3p	AGCTACATTGTCTGCTGGGT
ssc-miR-222	CTACATCTGGCTACTGGGT
ssc-miR-23a	TCACATTGCCAGGGATTTCC
ssc-miR-22-3p	AGCTGCCAGTTGAAGAACTGT
ssc-miR-7-3p	CAACAAATCACAGTCTGCC
U6-F	GGAACGATACAGAGAAGATTAGC
U6-R	TGGAACGCTTCACGAATTTGCG

## Data Availability

The data used in this study are presented in this manuscript and the Appendix A, and the sequencing data were submitted to the SRA database in NCBI (bioproject accession number: PRJNA940673). The link to the data is as follows: https://dataview.ncbi.nlm.nih.gov/object/PRJNA940673?reviewer=j14pq0i2tpskcak95htmbclass (accessed on 3 March 2023).

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
