# Peer review of "Breed-Related Differential microRNA Expression and Analysis of Colostrum and Mature Milk Exosomes in Bamei and Landrace Pigs"

_ijms, 2024, doi:10.3390/ijms25010667_

Round 1
Reviewer 1 Report
Comments and Suggestions for Authors
The manuscript "Breed-related differential miRNAs expression and analysis of colostrum and mature milk exosomes in Bamei and Landrace pigs" submitted for review presents the molecular analysis of exosomes occurring in pig colostrum and milk. In the light of available knowledge, exosomes are responsible for the transfer of intercellular information and take an active part in shaping immunity in newborn organisms. The authors showed that animal breed influences the number and effects of exosomes found in milk. This is due to large differences in the resistance and development speed of piglets of compared breeds. The authors created a rich molecular library of exosomes isolated from colostrum and milk. It seems interesting whether feeding them with colostrum and/or milk from breeds producing milk richer in exosomes can improve the immunity of piglets?
The only doubts I have are the description of Fig. 1. – do the presented exosomes come from milk or colostrum? While the shape or size of exosomes may not change regardless of their origin from colostrum or milk (although we are not sure), their concentration probably does (the authors themselves point out this regularity in the text).
Author Response
Dear reviewers, we have uploaded the response letter to the attachment, please download it.

Reviewer 2 Report
Comments and Suggestions for Authors
Author Response

(The authors gave the same response as above.)

Reviewer 3 Report
Comments and Suggestions for Authors
The article describes the differential content of milk exosome miRNA depending on pig breed (one is a local breed, the second a selected commercial breed) and the lactation period: colostrum and mature milk.
The results are not well discussed, and some of them are even not discussed at all, as for example the differences in exosome characterization. There is no information about exosomes, Are exosomes from mature milk or colostrum?
Sequencing results are widely explained, but finally the differences between breeds and between milk or colostrum are not discussed. The authors must make an effort in discussing all the results, and give some conclusions of their work.
Materials and methods must also be improved.
Author Response

(The authors gave the same response as above.)

Reviewer 4 Report
Comments and Suggestions for Authors
In the manuscript the authors reported about the differences in milk exosomes related to breed and milk types. Exosomes were isolated from colostrum and mature milk from domestic Bamei pigs and foreign Landrace pigs, authors identified a total of 1,081 known and 2,311 novel miRNAs in pig milk exosomes from Bamei and Landrace pigs.
The topic is interesting. There are some major issues that the authors may want to consider before publication.
I have some major comments:
1. The authors do not provide evidence that exosomes have been isolated. To confirm the exosomal nature of the isolated vesicles, the International Society for the Study of Extracellular Vesicles recommended the identification of specific exosomal membrane proteins - tetraspanins CD9, CD63 and CD81, using Western blotting, flow cytometry or immunoelectron microscopy [J Extracell Vesicles. 2018, V. 23, P. 7(1) : 1535750. doi: 10.1080/20013078.2018].
2. Please describe in more detail the characterization of exosomes by TEM and Nanoflow detector in chapters «4.3. Characterization of milk exosomes» and «2.1. Exosome characterization».
3. Since exosomes have been isolated from colostrum and mature milk of different pigs - domestic Bamei pigs and foreign Landrace pigs, please include TEM images and the data about average particle size for exosome preparation from different sources in chapter «2.1. Exosome characterization». Were there morphological differences between exosomes isolated from colostrum and mature milk?
4. Figures 1 A and B show electronic photographs of exosome preparations from Bamei and Landrace pigs. However, Figure 1B shows not only vesicles/exosomes, but also clusters of amorphous matter, which may be a protein cluster. The authors found 2,311 microRNAs in these milk exosomes preparations. The question arises: are these microRNAs contained in exosomes or in clusters of amorphous matter? In my opinion the authors should work on this issue more carefully.
5. Have exosome preparations been treated with RNase, to remove impurity RNA which was bind with protein clusters? The authors may consider the paper Valadi H et al. [Nat Cell Biol. 2007, V. 9(6), P. 654-659. doi: 10.1038/ncb1596]. In this paper exosome samples were treatment by trypsin to remove any non-exosomal macromolecules and RNase for removal of non-exosomal RNA.
Author Response

(The authors gave the same response as above.)

Round 2
Reviewer 3 Report
Comments and Suggestions for Authors
The paper is interesting, there is a lot of effort in presenting the miRNA results and their possible function. However, these enormous results are not well enough discussed to have an article for publication. An effort is needed to pool all the results in order to discuss them and give the article substance.
For example in the case of exosome characterisation. Do we know if the size of exosomes is statistically different between breeds or time of lactation?
In lines 100-101, why does the size and concentration of the exosomes allow us to know that the quality of the RNA is good enough to do the sequencing analysis?
It is also important to discuss why breed may condition miRNA expression. Is genetic selection acting to decrease the content of exosomes or certain miRNAs?
In lines 292-293, the authors say that Landrace has a higher number of exosomes but we do not have a statistic to confirm this. They say that one possible cause is the differences in the developmental stages between the breeds, but isn't this more likely to be the consequence?
Conclusions need to be improved
Author Response
Dear reviewers,
Thank you very much for taking your valuable time to review our manuscript, we have uploaded the response letter as an attachment, please download it!
Best regards
Xiaoyu Huang

Reviewer 4 Report
Comments and Suggestions for Authors
The authors have addressed all the reviewers' comments and therefore I recommend this manuscript for publication.
Author Response
Dear reviewer,
Thank you very much for agreeing to publish this research article, thanks again for taking your valuable time to review our manuscript, and have a nice life!
Best regards,
Xiaoyu Huang
Round 3
Reviewer 3 Report
Comments and Suggestions for Authors
The manuscript has been improved, even if all the results are not sufficiently discussed.